**Data Availability Statement:** As Ethiopian demographic and health survey is part of demographic and health survey (DHS), it is publicly

# Geographical variation and factors associated with unsafe child stool disposal in Ethiopia: A spatial and multilevel analysis

Biniyam Sahiledengle[1]*, Zinash Teferu[1], Yohannes Tekalegn[1], Tadesse Awoke[2], Demisu Zenbaba[1], Kebebe Bekele[3], Abdi Tesemma[3], Fikadu Seyoum[4], Demelash Woldeyohannes[5]

1 Department of Public Health, Madda Walabu University Goba Referral Hospital, Bale-Goba, Ethiopia, 2 Department of Epidemiology and Biostatistics, College of Medicine and Health Sciences, Institute of Public Health, University of Gondar, Gondar, Ethiopia, 3 Department of Surgery, Madda Walabu University Goba Referral Hospital, Bale-Goba, Ethiopia, 4 Department of Pediatrics, Madda Walabu University Goba Referral Hospital, Bale-Goba, Ethiopia, 5 Department of Public Health, Wachemo University, Hosaena, Ethiopia

* biniyam.sahiledengle@gmail.com

## Abstract

### Background

Unsafe disposal of children's stool makes children susceptible to fecal-oral diseases and children remain vulnerable till the stools of all children are disposed of safely. There is a paucity of data on spatial distribution and factors associated with unsafe child stool disposal in Ethiopia. Previous estimates, however, do not include information regarding individual and community-level factors associated with unsafe child stool disposal. Hence, the current study aimed (i) to explore the spatial distribution and (ii) to identify factors associated with unsafe child stool disposal in Ethiopia.

### Methods

A secondary data analysis was conducted using the recent 2016 Ethiopian demographic and health survey data. A total of 4145 children aged 0–23 months with their mother were included in this analysis. The Getis-Ord spatial statistical tool was used to identify high and low hotspots areas of unsafe child stool disposal. The Bernoulli model was applied using Kilduff SaTScan version 9.6 software to identify significant spatial clusters. A multilevel multivariable logistic regression model was fitted to identify factors associated with unsafe child stool disposal.

### Results

Unsafe child stool disposal was spatially clustered in Ethiopia (Moran's Index = 0.211, p-value< 0.0001), and significant spatial SaTScan clusters of areas with a high rate of unsafe child stool disposal were detected. The most likely primary SaTScan cluster was detected in Tigray, Amhara, Afar (north), and Benishangul-Gumuz (north) regions (LLR: 41.62, p<0.0001). Unsafe child stool disposal is more prevalent among households that had

available data. The data we used which is the '2016 Ethiopian Demographic and Health Survey' were obtained from the DHS program (www. dhsprogram.com), but the 'Dataset Terms of Use' do not permit us to distribute this data as per data access instructions (http://dhsprogram.com/data/ Access-Instructions.cfm). Any researcher can access data after becoming an Authorized user. To get access for the dataset researchers must first be a registered user of the website (www. dhsprogram.com), and access permission has been provided, users may download the 2016 Ethiopian Demographic and Health Survey. In addition, the shape file of the map was freely available from https://africaopendata.org/dataset/ ethiopia-shapefiles.

**Funding:** The author(s) received no specific funding for this work.

**Competing interests:** The authors have declared that no competing interests exist.

unimproved toilet facility (AOR = 1.54, 95%CI: 1.17–2.02) and those with high community poorer level (AOR: 1.74, 95%CI: 1.23–2.46). Higher prevalence of unsafe child stool disposal was also found in households with poor wealth quintiles. Children belong to agrarian regions (AOR: 0.62, 95%CI 0.42–0.91), children 6–11 months of age (AOR: 0.65, 95%CI: 0.52–0.83), 12–17 months of age (AOR: 0.68, 95%CI: 0.54–0.86), and 18–23 months of age (AOR: 0.58, 95%CI: 0.45–0.75) had lower odds of unsafe child stool disposal.

## Conclusions

Unsafe child stool disposal was spatially clustered. Higher odds of unsafe child stool disposal were found in households with high community poverty level, poor, unimproved toilet facility, and with the youngest children. Hence, the health authorities could tailor effective child stool management programs to mitigate the inequalities identified in this study. It is also better to consider child stool management intervention in existing sanitation activities considering the identified factors.

## Background

Disposal of child stool within the open fields, garbages, drainages, and burying in soils are considered unsafe as it exposes vulnerable children who interact with such to many fecal-oral diseases [1, 2]. Unsafe disposal of children's stool makes children vulnerable to numerous fecal-oral infections [1, 3–5]. A systematic review by Gil et al. detailed that unsafe child stool disposal is associated with a 23% increase in the risk of diarrheal infections in children [6]. Another review showed that the disposal of a child's feces into a toilet decreases the chances of diarrhea by almost 25% in children under five years of age [7]. Furthermore, there are established evidences on the effect of young children's stool disposal and increased risk of stunting [8, 9].

In Ethiopia, hygienic child stool management could be a tremendous challenge and putting the nation among the most noticeably awful third of 38 African nations for the percentage of children whose feces are safely disposed of according to the Multiple Indicator Cluster Survey (MICS) [2]. According to a later pooled information from the Ethiopian DHS (2000–2016), 77 percent of children's feces disposed of unsafely [10]. In Ethiopia, diarrhea is the leading cause of illness and contributor to deaths for under-five children [11]. Based on the WHO/CHERG estimates, diarrhea contributes to more than one in each ten (13%) child deaths in Ethiopia [12]. It isn't astonishing that diarrhea could be a significant contributor to under-five mortality, where open defecation is still exceptionally tall, get to progressed toilets is very low, and the handling of child feces is very poor [10, 11].

Previous studies has identified multiple factors that contribute to the occurrence of unsafe child stool disposal [10, 13–20]. Socioeconomic and demographic factors (such as household wealth index, the age of the child, sex of the child, age of the mother, maternal educational status, and place of residence) [10, 13, 14, 21, 22], and access to sanitation facility [10, 13, 14, 17, 19] were related with unsafe child stool disposal practices.

So far, few studies were conducted on child stool disposal in Ethiopia [10, 13, 14], and previous estimates, however, identified the determinants of child stool disposal using a standard logistic regression model that could ignore community-level variables, which may nullify or weaken the relation of the distal community-level factors [10, 13, 14]. Hence, a multilevel regression model is required, which considers the hierarchal and cluster nature of the

Ethiopian Demographic and Health Survey (EDHS) data and enhances the accuracy of estimates. And to date, studies on child stool disposal in Ethiopia have not assessed the spatial distribution of child stool disposal. Therefore, the current study aimed (i) to explore the spatial distribution and (ii) to identify factors associated with unsafe child stool disposal in Ethiopia using a multilevel regression model.

## Methods

### Study settings

Ethiopia is situated in the Horn of Africa between 3 and 15 degrees north latitude and 33 and 48 degrees east longitude. The total area of the country is about 1.1 million square kilometers and Djibouti, Eritrea, Sudan, Kenya, and Somalia border it. Contextually, the country is categorized as agrarian, pastoralists, and city-based population. Its peoples altogether speak over 80 different languages [23]. It has a total of 116,992,450 populations, of which 24,463,423 (21.3%) of the population is urban [24]. The health system of the Federal Democratic Republic of Ethiopia has a three-tier health-delivery service system. The primary level consists of primary healthcare units (health posts and health centers) and primary hospitals; secondary level services are provided by general hospitals, and tertiary services by specialized hospitals.

### Study design, data source, and extraction

This was a cross-sectional study that used data sets of the Ethiopia Demographic and Health Survey (EDHS) conducted in the year 2016. We analyzed stool disposal of children under the age of 2 years from the 2016-EDHS. The EDHS-2016 data sets were downloaded in STATA format with permission from the Measure DHS website (http://www.dhsprogram.com). The EDHS-2016 is the recent survey implemented by the Central Statistical Agency (CSA). The EDHS survey was used a sample weight for a study population to represent results at residence, region, and country level. The EDHS used a stratified two-stage cluster sampling technique. A detailed description of the study technique, sampling procedure, and surveys utilized for data collection is provided elsewhere [11]. The Global Positioning System (GPS) reading was collected at the center of each cluster. For the purpose of ensuring respondents' confidentiality, GPS latitude/ longitude positions for all surveys were randomly displaced before public release. Geographic coordinate data (latitude and longitude) was accessed through the online request system after registration as an authorized user in DHS international (http://www.dhsprogram.com). The shapefile of the map of Ethiopia was accessed as an open-source without restriction from Open Africa 2016 (https://africaopendata.org/dataset/ethiopia-shapefiles). The present study included the youngest child under age 2 living with the mother. A weighted total sample of 4145 children aged 0–23 months with their mother was included in the final analysis.

### Study variables

**Outcome variable.**   The outcome variable for this study was child stool disposal, which is dichotomized as unsafe and safe. A child's stool was considered to be disposed of "safely" when the child used latrine/ toilet or child's stool was put/rinsed into a toilet/latrine, whereas other methods were considered "unsafe". Therefore, we consider child stool disposal (unsafe = 1 or safe = 0) as outcome variable.

**Independent variables.**   The independent variables for this study were classified as individual and community level factors. The individual-level variables of this study were (age of the child, sex of the child, presence of diarrhea in the last two weeks, the source of drinking water, sanitation facilities, mother educational level, mother occupation, and household wealth

quintile). The contextual region, place of residence, and community poverty were identified as community-level variables (**Table 1**). The choice of independent variables was guided by already existing literatures [10, 13–15]. Community poverty level was generated by aggregating the individual characteristics in a cluster since EDHS did not collect data that can directly describe the characteristics of the clusters. Accordingly, community poverty level was an aggregate wealth index categorized as high or low, which is the proportion of women in the poorest and poorer quintile derived from data on wealth index which is categorized as low and high poverty community based on the median value. The interest of the current study was not in the regions delineated for administrative purposes, which might not necessarily be related to child stool disposal of the population. Accordingly, in the current study, the regions were categorized into agrarian, pastoralist, and city. The regions of Tigray, Amhara, Oromiya, SNNP, Gambella, and Benshangul Gumuz were categorized as agrarian. The Somali and Afar regions were categorized as the pastoralist region and the city administrations- Addis Ababa, Dire Dawa, and Harar were categorized as the city (**S1 File**).

## Data management and analysis

Secondary data obtained were imported from the Measure DHS website (http://www.dhsprogram.com) and analyzed using the STATA statistical software system package

**Table 1. Independent variables and categorization.**

| Individual-level factors | Category |
|---|---|
| **Child characteristics** | |
| Sex of the child | (1) male (2) female |
| Age of child (months) | (1) 0–5 (2) 6–11 (3) 12–17 (4) 18–23 |
| Diarrhea in the last two weeks | (1) yes (2) no |
| **Maternal/paternal/household** | |
| **Characteristics** | |
| Maternal age in years | (1) 15–24 (2) 25–34 (3) ≥ 35 |
| Educational level of mother | (1) no formal education (2) primary (3) secondary (4) higher |
| Mother's employment status | Categorized in to (1) not employed, or (2) employed |
| Number of under-five children | (1) ≤ 2 (2) ≥ 3 |
| Wealth index[a] | (1) (first quintile) (poorest) (2) (second quintile) (poorer); (3) (third quintile) (middle) (4) (fourth quintile) (richer) (5) (fifth quintile) (richest) |
| Source of drinking water[b] | (1) improved (2) unimproved |
| Toilet facility | (1) improved (2) unimproved |
| **Community-level factors** | |
| Place of residence | (1) urban (2) rural |
| Region | (1) agrarian (2) pastoralist (3) city |
| Community poverty | (1) high (2) low |

[a]Households are given scores based on the number and kinds of consumer goods they own, ranging from a television to a bicycle or car, in addition to housing characteristics such as source of drinking water, toilet facilities, and flooring materials. These scores are derived using principal component analysis. Household wealth index categorized in quintiles: poorest, poorer, middle, richer and richest.;
[b]Improved sources of drinking water include piped water, public taps, standpipes, tube wells, boreholes, protected dug wells and springs, and rainwater.

version 14.0 (StataCorp., College Station, TX, USA). A sampling weight was done to adjust for the non-proportional allocation of the sample to different regions and the possible differences in response rates. A detail explanation on weighting procedure has been sufficiently described in EDHS methodology [11]. In Ethiopian DHS data, children within a cluster are more similar to each other than between clusters. As such, a multilevel model is generally more appropriate than the standard regression model because it allows analysis based on hierarchical structure of variables. For this reason, a multilevel model was used to identify factors associated with unsafe child stool disposal. The data correlated, having intra-class correlation (ICC) = 39.6 and 29.6% for the null and final model, respectively, which shows the data were significantly clustered. As the response variable was dichotomous (safe, unsafe), multilevel binary logistic regression was fitted. The model goodness of fit was checked using deviance and Akakie Information Criteria (AIC). The model with the lowest deviance and AIC was chosen. The Proportional Change in Variance (PCV) was computed for each model with respect to the empty model to show the power of the factors in the model to explain unsafe child stool disposal. Accordingly, the PCV was calculated by the following formula [PCV = (Ve-Vmi)/Ve], where Ve is variance in unsafe child stool disposal in the empty model and Vmi is variance in successive models. [Median Odds Ratio (MOR) = $\sqrt{2 \times V \times 0.6745}$ ~ exp $(0.95\sqrt{V})$], where V is the estimated variance of clusters. The MOR measure is always greater than or equal to 1. If the MOR is 1, there is no variation between clusters. Variables with p-value < 0.25 in the bi-variable analysis were fitted in the multivariable model. Adjusted Odds Ratio (AOR) with a 95% Confidence Interval (CI) and p-value <0.05 in the multivariable model were used to declare significant association with unsafe child stool disposal. Variance inflator factor (VIF) was employed for checking multicollinearity among the independent variables.

## Spatial autocorrelation analysis

In this study, the spatial analysis was performed using the spatial statistics tool (ArcGIS Version 10.3; Redlands, California, United States). The spatial autocorrelation (Global Moran's I) statistic measures were used to evaluate whether unsafe child stool disposal was dispersed, clustered, or randomly distributed [25]. Spatial heterogeneity of high /low areas of unsafe child stool disposal was examined using the Getis-Ord Gi* statistics and associated Z-scores. Moreover, the spatial interpolation technique was applied (using the ordinary kriging interpolation technique) to predict the unsampled /unmeasured value from sampled measurements.

## Spatial scan statistical analysis

Spatial scan statistical analysis was employed to identify the geographical locations of statistically significant spatial clusters of unsafe child stool disposal in Ethiopia using SaTScan™ version 9.6 software. Unsafe child stool was taken as cases and those with safe child stool disposal as controls to fit the Bernoulli model [26]. The default maximum spatial cluster size of <50% of the population was used. A Likelihood ratio test statistic was used to determine whether the number of observed unsafe stool disposal cases within the potential cluster was significantly higher than the expected or not. Primary and secondary clusters were identified using p-values and log-likelihood ratio tests.

## Ethics approval

The analysis displayed in the paper is based on the Ethiopian Demographic Health Survey-2016 which is a publicly available dataset with no identifiable information on the study

members. The IRB-approved procedures for DHS public-use datasets do not in any way allow respondents, households, or sample communities to be identified. There are no names of individuals or household addresses in the data files. The geographic identifiers only go down to the regional level (where regions are typically very large geographical areas encompassing several states/provinces). Each EA (primary sampling unit) has a number in the data file, but their numbers do not have any labels to indicate their names or locations. The detail of the ethical issues has been published in the 2016 EDHS final report [11]. All the ethical concerns, including informed consent, are entirely followed in the EDHS-2016. Given these, no ethical approval or informed consent was required for the current study.

## Results

### Socio-demographic characteristics of participants

Table 2 lists the individual and community-level characteristics of the children included in this analysis. A total of 4,145 children aged 0–23 months with their mother were included in the final analysis. Of these, 2,164 (52.2%) were female with a mean age of 10.66 months (SD ± 0.11). The majority of 3,647 (88.0%) of the respondents were rural residents. About 2,500 (60.3%) of the children's mother had no formal education and about one-fifth were in the poorest wealth quintile (Table 2).

### Unsafe child stool disposal status

The prevalence of unsafe child stool disposal was 63.10% (95%CI: 59.5–66.6%). As shown in Table 3, binary multilevel logistic regression analysis was used to present unadjusted OR (95% CI) for individual and community level variables to identify factors associated with unsafe child stool disposal. Individual level characteristics such as sex of the child, age of the child, diarrhea conditions of the child, mother educational level, mother's employment status, household wealth index, toilet facility, and source of drinking water were significantly associated with unsafe child stool disposal at $p < 0.05$ (Table 3). All the community level characteristics (region, place of residence, and community poverty level) were found to be significantly associated with unsafe child stool disposal at $p < 0.05$ (Table 3).

### Spatial distribution of unsafe child stool disposal in Ethiopia

The analysis of spatial autocorrelation indicated that the spatial distribution of unsafe child stool disposal was clustered in Ethiopia. The Global Moran's I value 0.211 (p-value <0.0001) indicated that there was significant clustering of unsafe child stool disposal in Ethiopia (S1 Fig).

Hot-spot areas were found in Tigray (Central, and northeast), Amhara (Central, North, and Southeast), Afar (West, and South), Gambela (West), Oromia (South and East), North and Southeast parts of Somali regions, while cold-spot areas were found in SNNP (North, West, and East), Benishangul- Gumuz (Southwest), Addis Ababa, Harari and Dire Dawa (Fig 1).

Ordinary kriging interpolation analysis was conducted to predict child stool disposal in Ethiopia. High unsafe child stool disposal areas were found in Tigray, Amhara, Afar, Gambela, Southern Somali, and Southeastern parts of Oromia regions. In contrast, low unsafe child stool disposal areas were predicted in SNNP, Southern parts of Benishangul-Gumuz, Northern Somali, Western Oromia, and some parts of Amhara regions (Fig 2).

### Spatial scan statistical analysis

Table 4 show significant spatial clusters of unsafe child stool disposal in Ethiopia. Most likely (primary clusters) and secondary clusters of unsafe child stool disposal were identified. A total

**Table 2. Socio-demographic and socio-economic characteristics of study participants, EDHS 2016 (N = 4145).**

| Background characteristics | Weighted frequency (n) | % |
|---|---|---|
| **Individual-level factors** | | |
| **Sex of the child (n = 4144)[#]** | | |
| Male | 1980 | 47.8 |
| Female | 2164 | 52.2 |
| **Age of the child** | | |
| 0–5 months | 1059 | 25.6 |
| 6–11 months | 1085 | 26.2 |
| 12–17 months | 814 | 19.6 |
| 18–23 months | 1187 | 28.6 |
| **Diarrhea in the last two weeks (n = 4129)[#]** | | |
| Yes | 670 | 16.2 |
| No | 3459 | 83.8 |
| **Mother educational level** | | |
| No formal education | 2500 | 60.3 |
| Primary | 1279 | 30.9 |
| Secondary | 254 | 6.1 |
| Higher | 112 | 2.7 |
| **Mother's age (4143)[#]** | | |
| 15–24 | 1215 | 29.3 |
| 25–34 | 2105 | 50.8 |
| >34 | 823 | 19.9 |
| **Mother's employment status** | | |
| Not employed | 2439 | 58.85 |
| Employed | 1705 | 41.15 |
| **Number of under-five children** | | |
| 1–2 | 3458 | 83.43 |
| ≥ 3 | 687 | 16.57 |
| **Household wealth index** | | |
| Poorest | 975 | 23.5 |
| Poorer | 905 | 21.8 |
| Middle | 867 | 20.9 |
| Richer | 754 | 18.2 |
| Richest | 642 | 15.5 |
| **Toilet facility** | | |
| Improved [a] | 419 | 10.1 |
| Unimproved | 3726 | 89.9 |
| **Source of drinking water** | | |
| Improved [b] | 2330 | 56.2 |
| Unimproved | 1815 | 43.8 |
| **Community-level factors** | | |
| **Contextual region** | | |
| Agrarian | 3802 | 91.74 |
| Pastoralist | 210 | 3.19 |
| City | 132 | 5.07 |
| **Place of residence** | | |
| Urban | 498 | 12.0 |
| Rural | 3647 | 88.0 |

(*Continued*)

**Table 2.** (Continued)

| Background characteristics | Weighted frequency (n) | % |
|---|---|---|
| **Community poverty level** | | |
| High | 1647 | 39.74 |
| Low | 2498 | 60.26 |

[a]Facilities that would be considered improved if any of the following types: flush/pour flush toilets to piped sewer systems, septic tanks, and pit latrines; ventilated improved pit (VIP) latrines; pit latrines with slabs; and composting toilets. Other facilities including households with no facility or use bush/field were considered unimproved.

[b] Include piped water, public taps, standpipes, tube wells, boreholes, protected dug wells and springs, rainwater, and bottled water.

[#] The main reason why the total count fewer than 4145 was because of sampling weight and missing values. Total count 4145 unless otherwise given in brackets.

of 270 significant clusters were identified at which 201 were most likely (primary) and 69 secondary clusters. The primary clusters were located in Tigray, Amhara, and Afar regions. The primary clusters were centered at 13.351814 N, 38.353591 E with 471.07 km radius, a relative risk (RR) of 1.26, and the Log-Likelihood Ratio (LLR) of 41.62, at p<0.0001. The secondary clusters were typically located in the Gambela region and centered at 8.238420 N, 33.229506 E with 147.69 km radius, RR: 1.41, and LLR of 26.07 at p-value < 0.0001. It shows that children within the area had 1.41 times higher risk of unsafe child stool disposal than outside the area. The third clusters were located in Oromia (south) and Somali (southeast) regions and centered at 4.006703 N, 41.599741 E with 419.89 km radius, RR: 1.30 and LLR of 19.10 at p-value < 0.0001. The fourth clusters were located in Somali (north) regions and centered at 9.107168 N, 43.165843 E with 45.70 km radius, RR: 1.55, and LLR of 14.41 at p-value < 0.0001. The fifth clusters were typically located in Hareri regions and centered at 9.292185 N, 42.553365 E with 18.63 km radius, RR: 1.59, and LLR of 14.41 at p-value < 0.001. The six clusters were typically located in Oromia (northeast) regions and centered at 8.888553 N, 40.744565 E with 63.62 km radius, RR: 1.38, and LLR of 11.07 at p-value < 0.001 (**S2 File**). The bright pink colors indicate that the most statistically significant spatial windows contain primary clusters of unsafe child stool disposal in Ethiopia. There was high unsafe child stool disposal within the cluster than outside the cluster (**Fig 3**).

### Hotspot detection of prevalence of diarrhea and unsafe child stool disposal

In **Fig 4**, the exploratory visualization of the spatial distribution diarrhea and unsafe child stool disposal were indicated. The highest proportions of diarrhea were observed in SNNPR region (Southern Ethiopia), while the highest hotspot areas of unsafe child stool disposal were detected in Tigray region (Northern Ethiopia).

### Measures of variation (random-effects) and model fit statistics

**Table 5** shows the measures of variation (random intercept models) and model fit statistics. Model comparison was done using deviance. The comparison was done among model with no independent variables (the null model), model 1 (a model with only individual-level factors), model 2 (a model with only community-level factors), and model 3 (a model with both individual and community level independent variables simultaneously). A model with the lowest deviance (model 3) was selected. In the null model, significant variation in unsafe child stool disposal among mothers across communities was observed with an ICC of 39.61% justifying the use of multilevel analysis approach (i.e., variation in terms of unsafe safe child stool disposal

**Table 3. Binary multilevel logistic regression analysis to determine associated factors of unsafe child stool disposal in Ethiopia, EDHS 2016.**

| Background characteristics | Child stool disposal | | Crude OR (95%CI) | p-value |
|---|---|---|---|---|
| | Unsafe | Safe | | |
| **Individual-level factors** | | | | |
| **Sex of the child (n = 4144)** | | | | |
| Male | 1226 | 754 | 1 | |
| Female | 1272 | 892 | 0.84(0.72–0.98) | 0.037 |
| **Age of the child** | | | | |
| 0–5 months | 401 | 786 | 1 | |
| 6–11 months | 456 | 604 | 0.63 (0.51–0.79) | p<0.001 |
| 12–17 months | 444 | 641 | 0.69 (0.55–0.86) | 0.001 |
| 18–23 months | 346 | 467 | 0.54 (0.42–0.69) | p<0.001 |
| **Diarrhea in the last two weeks (n = 4129)** | | | | |
| Yes | 352 | 318 | 0.76(0.61–0.95) | 0.017 |
| No | 2133 | 1326 | 1 | |
| **Mother educational level** | | | | |
| No formal education | 1618 | 882 | 1 | |
| Primary | 715 | 564 | 0.66(0.54–0.80) | p<0.001 |
| Secondary | 128 | 126 | 0.69(0.50–0.94) | 0.018 |
| Higher | 38 | 74 | 0.42(0.27–0.64) | p<0.001 |
| **Mother's age (4143)** | | | | |
| 15–24 | 793 | 422 | 1 | |
| 25–34 | 1227 | 878 | 0.85(0.70–1.03) | 0.093 |
| >34 | 477 | 346 | 0.87(0.68–1.10) | 0.248 |
| **Mother's employment status** | | | | |
| Not employed | 1577 | 862 | 1 | |
| Employed | 1038 | 668 | 0.81(0.67–0.97) | 0.020 |
| **Number of under-five children** | | | | |
| 1–2 | 2167 | 1291 | 1 | |
| ≥ 3 | 448 | 239 | 1.07(0.85–1.33) | 0.558 |
| **Household wealth index** | | | | |
| Poorest | 771 | 204 | 8.02(6.01–10.73) | p<0.001 |
| Poorer | 598 | 307 | 3.85(2.84–5.22) | p<0.001 |
| Middle | 494 | 373 | 2.60(1.92–3.54) | p<0.001 |
| Richer | 391 | 363 | 1.84(1.35–2.49) | p<0.001 |
| Richest | 243 | 399 | 1 | |
| **Toilet facility** | | | | |
| Improved | 198 | 221 | 1 | |
| Unimproved | 2300 | 1426 | 2.53(1.99–3.21) | p<0.001 |
| **Source of drinking water** | | | | |
| Improved | 1298 | 1032 | 1 | |
| Unimproved | 1200 | 615 | 1.76(1.42–2.16) | p<0.001 |
| **Community-level factors** | | | | |
| **Region** | | | | |
| Agrarian | 2387 | 1415 | 1.65(1.16–2.38) | 0.006 |
| Pastoralist | 159 | 51 | 4.61(2.91–7.30) | p<0.001 |
| City | 69 | 63 | 1 | |
| **Place of residence** | | | | |
| Urban | 193 | 305 | 1 | |

*(Continued)*

**Table 3.** (Continued)

| Background characteristics | Child stool disposal | | Crude OR (95%CI) | p-value |
|---|---|---|---|---|
| | **Unsafe** | **Safe** | | |
| Rural | 2305 | 1342 | 3.89(2.85–5.30) | p<0.001 |
| **Community poverty level** | | | | |
| High | 1374 | 1124 | 5.05(3.84–6.62) | p<0.001 |
| Low | 1241 | 406 | 1 | |

could be attributed to unobserved community characteristics). Furthermore, between-cluster variability declined over successive models, from 39.61% in the empty model to 29.62% in the combined model. In the final model (model 3), individual and community-level factors accounted for about 35.87% of the variation observed for unsafe child stool disposal.

### Factors associated with unsafe child stool disposal

The calculated value intra-cluster correlation (ICC) was 39.61%, which indicated that the assumption of independent observation was violated (Table 5). Thus, we used a multilevel

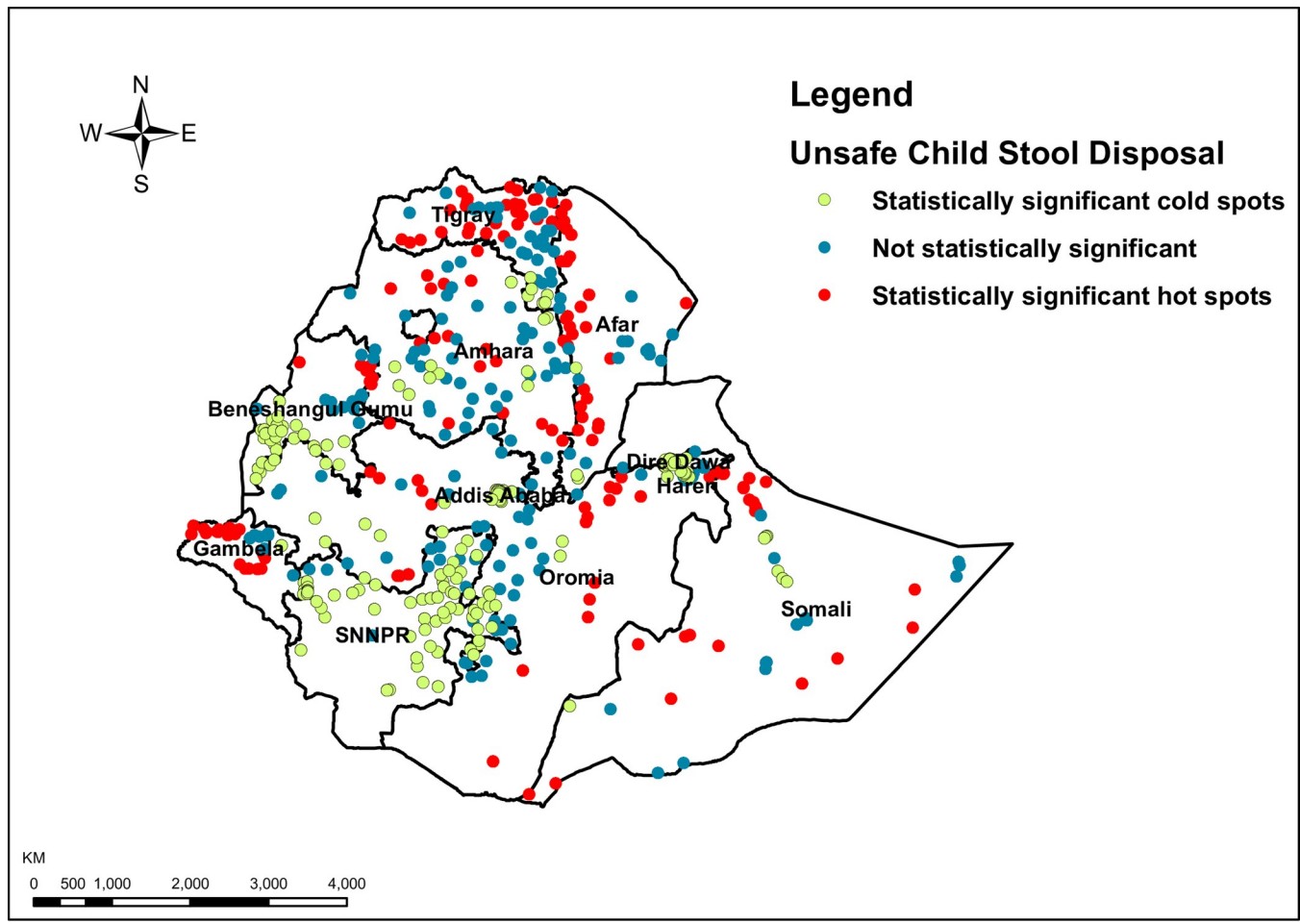

**Fig 1. Hotspot and cold spot analysis using Getis-Ord Gi statistics of unsafe child stool disposal in Ethiopia: A single dot on the map represents one enumeration area, EDHS 2016.** Source: Shapefile of the map was freely available from https://africaopendata.org/dataset/ethiopia-shapefiles.

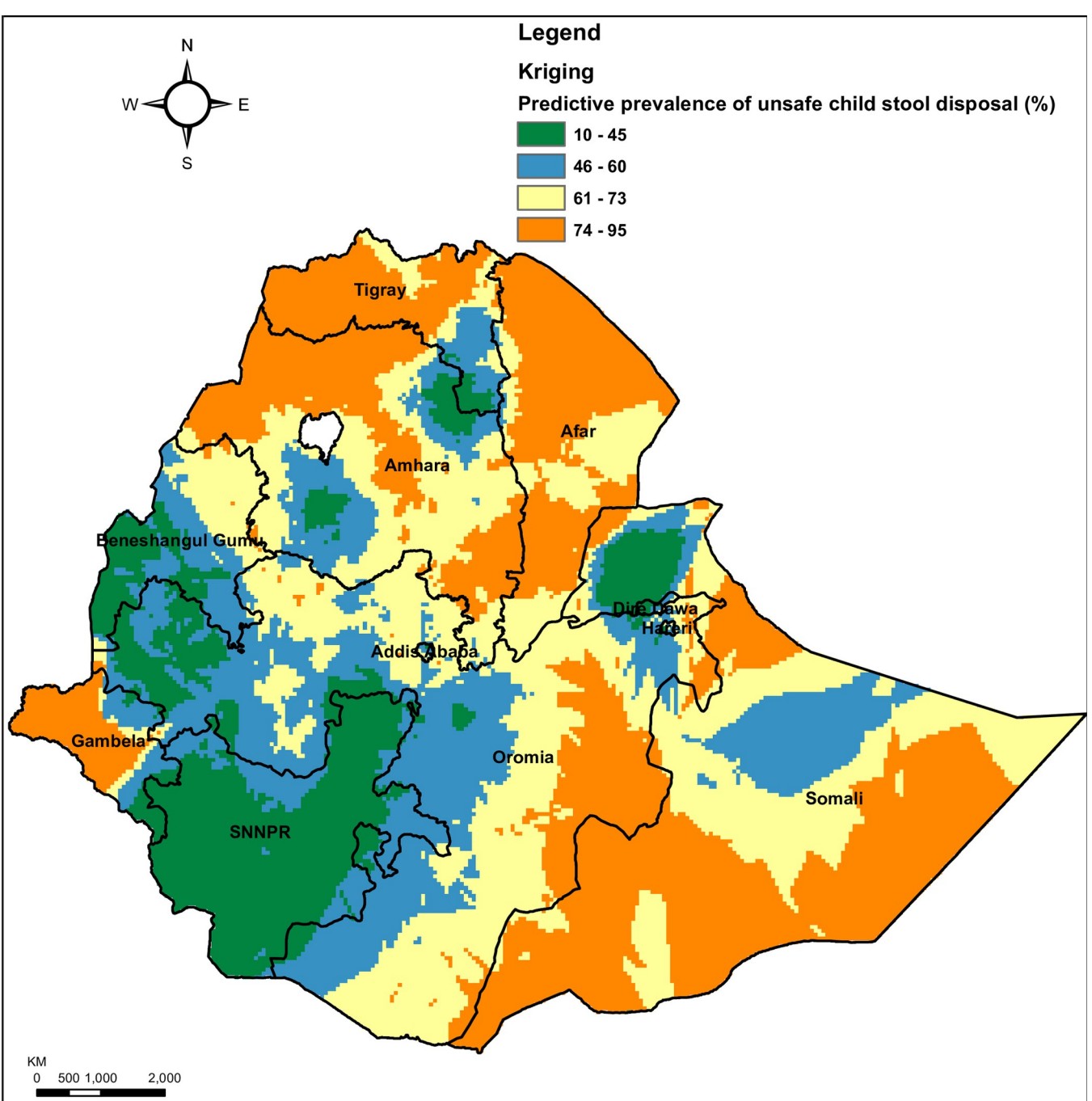

**Fig 2. Ordinary kriging interpolation of unsafe child stool disposal in Ethiopia, EDHS 2016.** Source: Shapefile of the map was freely available from https://africaopendata.org/dataset/ethiopia-shapefiles.

logistic regression model to account for the cluster effect. **Table 6** displays the adjusted estimate of the selected factors on unsafe child stool disposal in Ethiopia. Result from the final model (model 3) showed that, age of the child, wealth index, type of toilet facility, region, and community poverty level were significantly associated with unsafe child stool disposal. Women whose children were aged 6–11 months (AOR: 0.65, 95%CI: 0.52–0.83), 12–17

**Table 4. Significant and most likely clusters spatial clusters of unsafe child stool disposal in Ethiopia, EDHS 2016.**

| Clusters | Number of clusters detected | Population | Cases | RR | LLR | P-value |
|---|---|---|---|---|---|---|
| 1st most likely cluster | 201 | 1,272 | 931 | 1.26 | 41.62 | < 0.0001 |
| 2nd most likely cluster | 26 | 164 | 144 | 1.41 | 26.07 | < 0.0001 |
| 3rd most likely cluster | 26 | 256 | 206 | 1.30 | 19.10 | <0.0001 |
| 4th most likely cluster | 6 | 41 | 40 | 1.55 | 14.71 | <0.0001 |
| 5th most likely cluster | 3 | 27 | 27 | 1.59 | 12.41 | <0.001 |
| 6th most likely cluster | 8 | 81 | 70 | 1.38 | 11.07 | <0.001 |

months (AOR: 0.68, 95%CI: 0.54–0.86), and 18–23 months (AOR: 0.58, 95%CI: 0.45–0.75) were less likely to dispose of their children's stool unsafely compared with those whose children were aged 0–5 months. In this study, the odds of practicing unsafe disposal was reduced as household wealth quintiles increase. Children belonging to the poorest wealth quintiles had a four times higher chance of unsafe child stool disposal (AOR: 4.62, 95%CI: 2.98–7.16) than children belonging to the richest wealth quintiles. Similarly, children belonging to the poorer (AOR: 2.77, 95%CI: 1.82–4.23), middle (AOR: 2.13, 95%CI: 1.41–3.22), and richer wealth quintiles (AOR: 1.56, 95%CI: 1.05–2.32) had higher odds of unsafe child stool disposal than children belonging to richest wealth quintiles. Children belong to households who had unimproved toilet facilities were about 54% (AOR: 1.54, 95%CI: 1.17–2.02) more likely to had unsafe child stool disposal than children in households with improved toilet facilities. The likelihood of unsafe child stool disposal among children belong to agrarian regions was about 38% (AOR: 0.62, 95%CI 0.42–0.91) lower compared with city dwellers. Unsafe child stool disposal is more prevalent among households that are high community poorer level (AOR: 1.74, 95% CI: 1.23–2.46) than those with younger children live in low community poverty level (**Table 6**).

## Discussion

This study was aimed to explore geographical variation and identify the determinants of unsafe child stool disposal in Ethiopia. Our result indicated that unsafe child stool disposal was found to be a spatial problem in Ethiopia. Multilevel multivariable logistic regression analyses showed that individual level (the age of the child, wealth index, and types of toilet facilities were associated with unsafe stool disposal) and community-level factors (region and community poverty) were associated with unsafe child stool disposal in Ethiopia.

In Ethiopia, the proportion of unsafe child stool disposal was 63.10%; the highest proportion was detailed in rural areas (p<0.001). This finding was consistent with a cross-sectional study conducted in Ethiopia which found that 67% of households reported unsafe child stool disposal [14]. The present high unsafe child stool disposal noted in this study could be attributed to poor access to toilet facilities in Ethiopia, as having the toilet facilities is important to promote safe child stool disposal. According to the recent EDHS report, one in three households have no toilet facility and children's stools are more likely to be disposed of unsafely in households that use open defecation and have no toilet facility [11]. Relatively higher unsafe child stool disposal was also reported from studies done in India 72.4%-79% [3, 21] and Bangladesh 80%-84% [20, 27, 28]. The inconsistency between studies maybe due to the fact that in the aforementioned studies the operational definition utilized in classifying unsafe child stool disposal incorporates buried as safe child feces disposal [20]. The other possible reasons for this disparity may be due to study participants (i.e., previous studies include under-5 children in their analysis while we included the youngest child under age two).

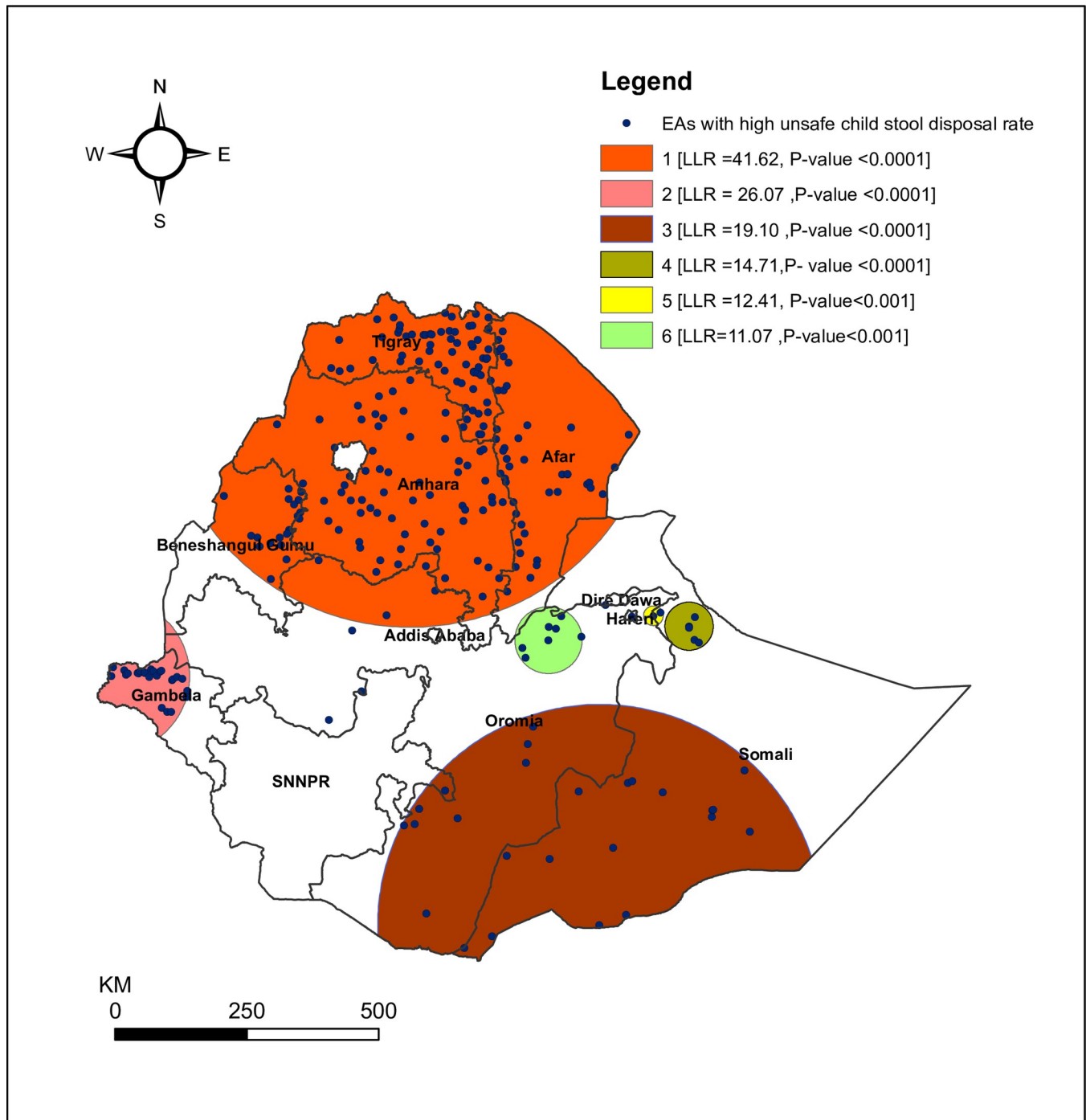

**Fig 3. The spatial clustering of areas with high unsafe child stool disposal in Ethiopia, EDHS 2016.** Source: Shapefile of the map was freely available from
https://africaopendata.org/dataset/ethiopia-shapefiles.

In the global spatial autocorrelation analysis of this study, a clustering pattern of unsafe child stool disposal across the study areas was observed (Global Moran's I = 0.211, p-value< 0.0001). This indicates that unsafe child stool disposal in Ethiopia was aggregated in specific areas. Accordingly, the hot-spot areas were found in Tigray (Central, and northeast), Afar

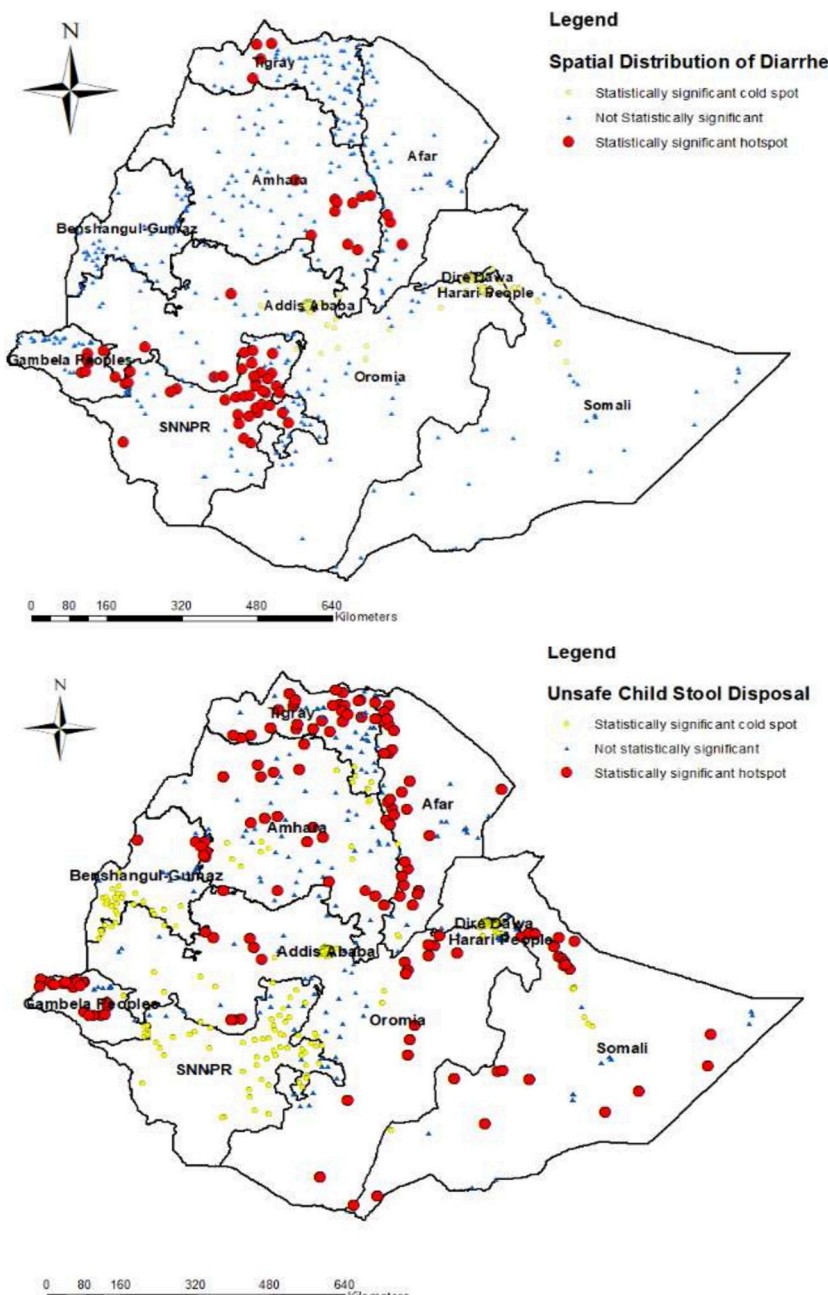

**Fig 4. The exploratory visualization of the spatial distribution diarrhea and unsafe child stool disposal.** (a) map at upper section showed the spatial distribution diarrhea (b) map at lower section showed the spatial distribution of unsafe child stool disposal, EDHS 2016. Source: Shapefile of the map was freely available from https://africaopendata.org/dataset/ethiopia-shapefiles.

(West, and South), Amhara (Central, North, and Southeast), Gambela (West), Oromia (East), North and some parts of Somali regions. The possible explanation for geographic variation in the prevalence of unsafe child stool disposal might be due to high open defecation practice in these identified hot spot areas. According to EDHS-2016 report, one in three households in Ethiopia have no toilet facility (39% in rural areas and 7% in urban areas) and open defecation

**Table 5. Measures of variation (random intercept models) and model fit statistics for unsafe child stool disposal in Ethiopia, EDHS 2016.**

| Individual- and community-level characteristics | Null model | Model 1[b] | Model 2[c] | Model 3[d] |
|---|---|---|---|---|
| | (Empty model) | Individual-level variables | Community-level variables | Individual- and community-level variables |
| **Random effect** | | | | |
| Community-level variance (SE) | 2.155(0.082)*** | 1.394(0.075)*** | 1.442(0.074)*** | 1.382(0.074)*** |
| ICC (%) | 39.61% | 29.78% | 30.51% | 29.62% |
| MOR[e] | 4.03 | 3.06 | 3.12 | 3.05 |
| PCV (%) | Reference | 35.31 | 33.08 | 35.87 |
| **Model fit statistics** | | | | |
| AIC | 4608.483 | 4357.293 | 4456.667 | 4344.191 |
| BIC | 4621.028 | 4476.419 | 4494.301 | 4488.395 |
| DIC(-2Log-likelihood) | 4604.484 | 4319.292 | 4444.668 | 4358.190 |

SE Standard Error; ICC: Intra-class Correlation Coefficient; MOR: Median Odds Ratio; PCV: Proportional Change in Variance; AIC: Akaike's Information Criterion;

BIC: Bayesian information Criteria; DIC: Deviance Information Criterion.

[a]Null model is an empty model, a baseline model without any explanatory variable.

[b]Model 1 is adjusted for individual-level factors.

[c]Model 2 is adjusted for community-level factors.

[d]Model 3 is the final model adjusted for both individual and community-level factors.

[e]Increased risk (in median) that one would have if moving to a neighborhood/cluster with a higher risk.

***P-value < 0.001.

was practiced in 32.9% of the households (37.7% in rural areas and 6.8% in urban areas). Closer looks in these hot spot areas showed that unsafe child stool disposal is relatively aggregated in rural areas. Consistent with this affirmation, unsafe child feces disposal is more prevalent among households that defecate in the open and those in rural areas; over three fourth of the rural households in Ethiopia (81.2%) had unsafe child feces disposal while that is true only for (45.8%) of the urban households [11]. In the community-level factors (model 2), our finding also suggested that the odds of unsafe child stool disposal were two times higher among children residing in rural areas. Additionally, the high proportion of unsafe child stool disposal in this area might be due to disparity in access to improved sanitation facilities.

Consistent with previous studies conducted in Ethiopia [10], Malawi [15], and Bangladesh [20, 29], women with older children were less likely to have unsafe child stool disposal compared with those with young children. This association can be explained by the fact that children being more likely to utilize a toilet themselves as they get older [2, 15, 30]. Moreover, the increased likelihood of unsafe child stool disposal among households with younger children may be due to the widespread false beliefs in the community that the stool of young children is considered harmless [2, 17].

Our study found that children belonging to the poorest and poorer wealth quintiles had a higher odds of unsafe child stool disposal than children in households with the richest wealth quintiles. This finding was consistent with other related studies [10, 13–15, 19]. In connection, there is also evidence in the current study; unsafe child feces disposal is more prevalent among households with high community poverty levels. In this study, children belong to households that had unimproved toilet facilities were more likely to had unsafe child stool disposal. These finding was consistent with some of previously conducted studies in Ethiopia [10, 14] and South Africa [31].

At the community level, children belong to agrarian regions (like SNNP and Beneshangul Gumuz regions) were less likely to have unsafe child stool disposal than city dwellers. This finding highlighted the require for solid sanitation programs in the city administration in

**Table 6. Multivariable multilevel logistic regression analysis to determine associated factors of unsafe child stool disposal in Ethiopia, EDHS 2016.**

| Background characteristics | Null model | Model 1 | Model 2 | Model 3 |
|---|---|---|---|---|
| | (Empty model) | Individual-level variables | Community-level variables | Individual- and community-level variables |
| | | AOR (95% CI) | AOR (95% CI) | AOR (95% CI) |
| **Individual-level factors** | | | | |
| **Sex of the child (n = 4144)** | | | | |
| Male | | 1 | | 1 |
| Female | | 0.86(0.73–1.01) | | 0.86(0.73–1.02) |
| **Age of the child** | | | | |
| 0–5 months | | 1 | | 1 |
| 6–11 months | | 0.66(0.52–0.83)*** | | 0.65(0.52–0.83)*** |
| 12–17 months | | 0.68(0.54–0.86)** | | 0.68(0.54–0.86)** |
| 18–23 months | | 0.58(0.45–0.74)*** | | 0.58(0.45–0.75)*** |
| **Diarrhea in the last two weeks (n = 4129)** | | | | |
| Yes | | 0.79(0.63–1.01) | | 0.82(0.65–1.03) |
| No | | 1 | | 1 |
| **Mother educational level** | | | | |
| No formal education | | 1 | | 1 |
| Primary | | 0.81(0.66–1.01) | | 0.85(0.68–1.05) |
| Secondary | | 1.16(0.83–1.62) | | 1.23(0.87–1.73) |
| Higher | | 0.97(0.62–1.52) | | 0.99(0.63–1.56) |
| **Mother's age (4143)** | | | | |
| 15–24 | | 1 | | 1 |
| 25–34 | | 0.86(0.70–1.05) | | 0.88(0.72–1.07) |
| >34 | | 0.82 (0.63–1.06) | | 0.85(0.65–1.10) |
| **Mother's employment status** | | | | |
| Not employed | | 1 | | 1 |
| Employed | | 0.92(0.77–1.10) | | 0.95(0.79–1.14) |
| **Household wealth index** | | | | |
| Poorest | | 6.35(4.49–8.99)*** | | 4.62(2.98–7.16)*** |
| Poorer | | 3.28(2.32–4.63)*** | | 2.77(1.82–4.23)*** |
| Middle | | 2.30(1.63–3.24)*** | | 2.13(1.41–3.22)*** |
| Richer | | 1.60(1.15–2.24)** | | 1.56(1.05–2.32)** |
| Richest | | 1 | | 1 |
| **Toilet facility** | | | | |
| Improved* | | 1 | | 1 |
| Unimproved | | 1.41(1.08–1.83)** | | 1.54(1.17–2.02)** |
| **Source of drinking water** | | | | |
| Improved** | | 1 | | 1 |
| Unimproved | | 1.17(0.95–1.44) | | 1.14(0.92–1.41) |
| **Community level factors** | | | | |
| **Region** | | | | |
| Agrarian | | | 0.57(0.52–1.12) | 0.62(0.42–0.91)** |
| Pastoralist | | | 1.34(0.84–2.13) | 0.87(0.54–1.40) |
| City | | | 1 | 1 |
| **Place of residence** | | | | |
| Urban | | | 1 | 1 |
| Rural | | | 2.17(1.49–3.16)*** | 1.08(0.68–1.72) |
| **Community poverty level** | | | | |

*(Continued)*

**Table 6.** (Continued)

| Background characteristics | Null model | Model 1 | Model 2 | Model 3 |
|---|---|---|---|---|
| | (Empty model) | Individual-level variables | Community-level variables | Individual- and community-level variables |
| | | AOR (95% CI) | AOR (95% CI) | AOR (95% CI) |
| High | | | 3.22(2.33–4.43)*** | 1.74(1.23–2.46)** |
| Low | | | 1 | 1 |

***p-value < 0.001;

**p-value< 0.05; AOR: Adjusted Odds Ratio.

Ethiopia. So far, the largest Community-Led Total Sanitation and Hygiene (CLTSH) endeavors to end open defecation have basically focused on rural communities in Ethiopia, with only a limited focus on the management of child stool among city dwellers [32, 33]. As a result, child feces management ought to be promoted among city dwellers within the country.

## Conclusion

This study showed that unsafe child stool disposal had spatial variability across survey clusters and regions; it was higher in the northern part of the country. Both the individual-level characteristics (child's age, wealth index, types of toilet facility) and community-level characteristics (region and community poverty) were statistically significant predictors of unsafe child stool disposal. Hence, the health authorities could tailor effective child stool management programs to mitigate the inequalities identified in this study. It is also better to consider child stool management intervention in existing sanitation activities.

## Limitations

Although used nationally representative data that can enhance the generalisability of the findings and a multilevel logistic regression model that accounts for the correlated nature of EDHS data. The present study has several limitations. First, due to the secondary nature of the data, the present study was limited by unmeasured confounders such as mother knowledge towards child stool disposal and other community-level factors such as social and cultural norms towards child feces management. Second, self-reported practices can be subject to bias that might underestimate true levels by underreporting socially undesirable behaviors. Additionally, EDHS self-reported child stool disposal practices have not been validated with objective measurements such as spot check observations. Third, the cross-sectional nature of the survey is not appropriate to estimate the cause and effect relationship between independent variables and unsafe child stool disposal.

## Supporting information

**S1 File. Basis for categorizing independent variables.**
(DOCX)

**S2 File. Significant spatial clusters of unsafe child stool disposal in Ethiopia, enumeration areas(clusters) detected and there coordinates/radius.**
(DOCX)

**S1 Fig. The global spatial autocorrelation based on feature locations and attribute values of unsafe child stool disposal in Ethiopia, EDHS 2016.**
(PDF)

## Author Contributions

**Conceptualization:** Biniyam Sahiledengle.

**Data curation:** Biniyam Sahiledengle.

**Formal analysis:** Biniyam Sahiledengle, Zinash Teferu.

**Methodology:** Biniyam Sahiledengle, Zinash Teferu, Yohannes Tekalegn.

**Software:** Biniyam Sahiledengle.

**Validation:** Biniyam Sahiledengle, Zinash Teferu, Yohannes Tekalegn, Tadesse Awoke, Demisu Zenbaba, Kebebe Bekele, Abdi Tesemma, Fikadu Seyoum, Demelash Woldeyohannes.

**Visualization:** Zinash Teferu, Yohannes Tekalegn, Tadesse Awoke, Demisu Zenbaba, Kebebe Bekele, Abdi Tesemma, Fikadu Seyoum, Demelash Woldeyohannes.

**Writing – original draft:** Biniyam Sahiledengle.

**Writing – review & editing:** Zinash Teferu, Yohannes Tekalegn, Tadesse Awoke, Demisu Zenbaba, Kebebe Bekele, Abdi Tesemma, Fikadu Seyoum, Demelash Woldeyohannes.

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
