## [Decision Letter · Decision Letter 0]

28 Jan 2021

PONE-D-20-24200

Geographical variation and factors associated with unsafe child stool disposal in Ethiopia: A spatial and multilevel analysis

PLOS ONE

Dear Dr. Sahiledengle,

Thank you for submitting your manuscript to PLOS ONE. After careful consideration, we feel that it has merit but does not fully meet PLOS ONE’s publication criteria as it currently stands. Therefore, we invite you to submit a revised version of the manuscript that addresses the points raised during the review process.

The manuscript has been evaluated by three reviewers, and their comments are available below.

The reviewers have raised a number of concerns that need attention. They request additional information on methodological aspects of the study and the impact of potential confounding variables on the study results.  Furthermore, the reviewers feel that the tables presented in the results section could be improved.

Could you please carefully revise the manuscript to address all comments raised?

We look forward to receiving your revised manuscript.

Kind regards,

Lucinda Shen, MSc

Staff Editor

PLOS ONE

Journal Requirements:

https://bmcpublichealth.biomedcentral.com/articles/10.1186/s12889-016-3948-2

https://www.researchsquare.com/article/rs-3713/v2

https://www.mdpi.com/1660-4601/17/9/3084/htm

https://www.researchsquare.com/article/rs-637/v1

In your revision ensure you cite all your sources (including your own works), and quote or rephrase any duplicated text outside the methods section. Further consideration is dependent on these concerns being addressed.

6. We note that Figures 1, 2, 3 in your submission contain map images which may be copyrighted. All PLOS content is published under the Creative Commons Attribution License (CC BY 4.0), which means that the manuscript, images, and Supporting Information files will be freely available online, and any third party is permitted to access, download, copy, distribute, and use these materials in any way, even commercially, with proper attribution. For these reasons, we cannot publish previously copyrighted maps or satellite images created using proprietary data, such as Google software (Google Maps, Street View, and Earth). For more information, see our copyright guidelines: http://journals.plos.org/plosone/s/licenses-and-copyright.

6.1.    You may seek permission from the original copyright holder of Figures 1, 2, 3 to publish the content specifically under the CC BY 4.0 license. 

6.2.    If you are unable to obtain permission from the original copyright holder to publish these figures under the CC BY 4.0 license or if the copyright holder’s requirements are incompatible with the CC BY 4.0 license, please either i) remove the figure or ii) supply a replacement figure that complies with the CC BY 4.0 license. Please check copyright information on all replacement figures and update the figure caption with source information. If applicable, please specify in the figure caption text when a figure is similar but not identical to the original image and is therefore for illustrative purposes only.

Reviewers' comments:

Reviewer's Responses to Questions

**Comments to the Author**

1. Is the manuscript technically sound, and do the data support the conclusions?

Reviewer #1: Yes

Reviewer #2: Yes

Reviewer #3: Yes

2. Has the statistical analysis been performed appropriately and rigorously? 

Reviewer #1: Yes

Reviewer #2: Yes

Reviewer #3: Yes

3. Have the authors made all data underlying the findings in their manuscript fully available?

Reviewer #1: Yes

Reviewer #2: Yes

Reviewer #3: Yes

4. Is the manuscript presented in an intelligible fashion and written in standard English?

Reviewer #1: Yes

Reviewer #2: Yes

Reviewer #3: Yes

5. Review Comments to the Author

Reviewer #1: Geographical variation and factors associated with unsafe child stool disposal in Ethiopia: A spatial and multilevel analysis

Very well written manuscript. This is really a very important and timely work as in the LMICs still struggling with open defection and child feces disposal. Very few works done in this area and I really appreciate authors for this work.

I really like the way presented the findings as well as noticed the limitation of the study. However, few minor comments might improve the manuscript.

1. Author didn’t mention the name of the institution where received Institutional Review Board approval.

2. Vast majority of the mother 60.3% had no formal education, so it would be great to include whether any association of unsafe child feces disposal practices with mother’s education.

3. Having the toilet facilities is important to promote safe disposal, however if any household don’t have latrine how you considered in this analysis?

Reviewer #2: Sahiledengle and colleagues mapped the distribution and described the risk of unsafe disposal of child stool in Ethiopia. Here, the study importance, objectives, and methods are clearly stated.

Minor comments:

1) What would happen if you use a continuous age variable rather than an age group in the model? What are the basis for participants’ age categorization?

2) Include or explain in the supplementary or elsewhere the basis of categorizing the independent variables (e.g., wealth index, community poverty, etc.) to understand how you define the cut off and make sense to some of the nominal and dichotomous variables.

3) Table 4 is not informative.

4) I wonder if you could include or overlap data on the spatial prevalence of diarrhoea with spatial distribution of stool disposal. Their association is a strong basis for public health policy.

Reviewer #3: This is a very important and interesting research. The introduction is sound and well structured. The rationale and objectives of the study are clearly spelt out. The methodology section is well written but needs improvement. Authors need to consider the introduction of new subsections, and provide more clarity on the categorization of variables. The data management and analysis plan was described appropriately. The data analysis section is robust, and the authors have done a very great job at pulling all the data together. However, there are issues with results presentation and description within text for a particular table. In addition, concerns on why the authors decided to pull the administrative regions together under agarian, pastoralist and city categories before the multivariate logistic analysis needs to be rechecked. The discussion is sound and well written. More specific comments have been raised in the additional file attached and authors should consider working on them.

6. PLOS authors have the option to publish the peer review history of their article (what does this mean?). If published, this will include your full peer review and any attached files.

Reviewer #1: **Yes: **Md Mahbubur Rahman

Reviewer #2: **Yes: **Harvie P. Portugaliza

Reviewer #3: **Yes: **MOGAJI HAMMED OLADEJI

---

## [Author Response · Author response to Decision Letter 0]

13 Mar 2021

Response to Reviewers

PONE-D-20-24200

Title: Geographical variation and factors associated with unsafe child stool disposal in Ethiopia: A spatial and multilevel analysis

Authors response to editor comment

Dear Editor,

It is a prestigious opportunity for us to have constructive comments for the improvement of the current manuscript. We thank you for this opportunity and we are happy to submit a revised version of the manuscript that addresses the points raised by our respected reviewers. Respected editor, we also carefully considered and taken all your while we revise our manuscript. Please follow the point-by-point response to the editor comments which is listed below. Following your suggestion, we include a rebuttal letter that responds to each point raised by the editor and reviewers’ comment and we upload this letter as a separate file labeled 'Response to Reviewers'. We also upload a marked-up copy of our manuscript that highlights changes made to the original version as a separate file labeled 'Revised Manuscript with Track Changes'. Finally, we upload an unmarked version of our revised paper without tracked changes as a separate file labeled 'Manuscript'.

Editor comment and response 

Comment 1 

Journal Requirements:

Response 1

Thank you, our respected editor. As per your wise advice we revised the manuscript according to PLOS ONE's journal style requirements. Please see the revised manuscript. 

Comment 2

Response 2

Thank you, our respected editor. As per your wise advice we copyedit our manuscript by our colleague assistant professor Bruce John Edward Quisido a lecturer at Madda Walabu University Goba Referral hospital, in the Department of Nursing. All the affected revisions and corrections were highlighted with text highlighter color and uploaded as “Revised Manuscript with Track Changes”. Thank you.

Comment 3

https://bmcpublichealth.biomedcentral.com/articles/10.1186/s12889-016-3948-2

https://www.researchsquare.com/article/rs-3713/v2

https://www.mdpi.com/1660-4601/17/9/3084/htm

https://www.researchsquare.com/article/rs-637/v1

In your revision ensure you cite all your sources (including your own works), and quote or rephrase any duplicated text outside the methods section. Further consideration is dependent on these concerns being addressed.

 Response 3:

Thank you, our respected editor. This is always a learning opportunity for us, following your wise advice we carefully revised our manuscript to remove overlapping texts from previous publications, we rephrase and cite all sources in the revised manuscript. All the affected revisions and corrections were highlighted with text highlighter color and uploaded as “Revised Manuscript with Track Changes”. Thank you.

Comment 4:

Response 4:

Thank you, our respected editor for your assistant and support. In short, the present study used data from the Ethiopian Demographic and Health Survey (EDHS) which is available from the DHS program website after the study aim and objective were communicated. According to DHS dataset terms of use do not permit us to distribute this data as per data access instruction “The data must not be passed on to other researchers without the written consent of DHS.” For this reason, we include the following information under “Data Availability” statement. “As Ethiopian demographic and health survey is part of demographic and health survey (DHS), it is publicly available data. The data we used which is the ‘2016 Ethiopian Demographic and Health Survey’ were obtained from the DHS program (www.dhsprogram.com), but the ‘Dataset Terms of Use’ do not permit us to distribute this data as per data access instructions (http://dhsprogram.com/data/Access-Instructions.cfm). Any researcher can access data after becoming an Authorized user. To get access for the dataset researchers must first be a registered user of the website (www.dhsprogram.com), and access permission has been provided, users may download the 2016 Ethiopian Demographic and Health Survey. In addition, the shape file of the map was freely available from https://africaopendata.org/dataset/ethiopia-shapefiles.” Please see the revised manuscript. Thank you.

Comment 5

Response 5:

Thank you for your comment. As per your wise advice we moved the ethics statement in to the Method and Material section of the revised manuscript. Please see the revised manuscript. 

Comment 6:

6. We note that Figures 1, 2, 3 in your submission contain map images which may be copyrighted. All PLOS content is published under the Creative Commons Attribution License (CC BY 4.0), which means that the manuscript, images, and Supporting Information files will be freely available online, and any third party is permitted to access, download, copy, distribute, and use these materials in any way, even commercially, with proper attribution. For these reasons, we cannot publish previously copyrighted maps or satellite images created using proprietary data, such as Google software (Google Maps, Street View, and Earth). For more information, see our copyright guidelines: http://journals.plos.org/plosone/s/licenses-and-copyright.

6.1. You may seek permission from the original copyright holder of Figures 1, 2, 3 to publish the content specifically under the CC BY 4.0 license. 

6.2. If you are unable to obtain permission from the original copyright holder to publish these figures under the CC BY 4.0 license or if the copyright holder’s requirements are incompatible with the CC BY 4.0 license, please either i) remove the figure or ii) supply a replacement figure that complies with the CC BY 4.0 license. Please check copyright information on all replacement figures and update the figure caption with source information. If applicable, please specify in the figure caption text when a figure is similar but not identical to the original image and is therefore for illustrative purposes only.

Response 6:

Thank you, our respected editor, for your comment and wise advice. The figures (Fig 1, 2, and 3) included in the present are not copyrighted or previously copyrighted maps or satellite images. Yet all Figures included in this study were our study finding/results showing the spatial distribution of unsafe child stool disposal in Ethiopia and any third party is permitted to access, download, copy, and distribute (CC BY 4.0). In brief, the dataset we used to analyzed spatial data was obtained from the publicly available DHS program (www.dhsprogram.com) website, after the study objective was communicated and it can be published under the Creative Commons Attribution License (CC BY 4.0). In DHS surveys that collect GIS coordinates in the field, the coordinates are only for the enumeration area (EA) as a whole, and not for individual households, and the measured coordinates are randomly displaced within a large geographic area so that specific numeration areas cannot be identified. There are no names of individuals or household addresses in the data files. The geographic identifiers only go down to the regional level (where regions are typically very large geographical areas encompassing several states/provinces). Each enumeration area (Primary Sampling Unit) has a PSU number in the data file, but the PSU numbers do not have any labels to indicate their names or locations. The GPS reading was collected at the center of each cluster. For the purpose of insuring respondents’ confidentiality, GPS latitude/ longitude positions for all surveys were randomly displaced before public release and we used such dataset.(For your information, we attached the approval letter we obtained from DHS in supporting file). In addition, the shape files we used in this study was also obtained from publicly available Africa open data website https://africaopendata.org/dataset/ethiopia-shapefiles. Additionally, we mentioned the sources of dataset in each figure legends and in data availability statement. Please see revised manuscript. 

Response to reviewer’s comment 

For reviewer #1

Dear respected reviewer 1

Thank you for this learning opportunity. We are so happy to have your advice, comments and suggestion. We are also so glad to see our paper improved because of your comments and wise advice. Please follow a point-by-point response to the reviewer’s comment below. We used a “Text Highlight Yellow Color” for all affected revisions and corrections in the “'Revised Manuscript with Track Changes” File. If there are any comments that need corrections, we are so happy to learn from you. With all respect, thank you.

Reviewer #1: Geographical variation and factors associated with unsafe child stool disposal in Ethiopia: A spatial and multilevel analysis.

Very well written manuscript. This is really a very important and timely work as in the LMICs still struggling with open defection and child feces disposal. Very few works done in this area and I really appreciate authors for this work. I really like the way presented the findings as well as noticed the limitation of the study. However, few minor comments might improve the manuscript.

Response:

Thank you, Sir for this opportunity. 

Comment 1

1. Author didn’t mention the name of the institution where received Institutional Review Board approval.

Response 1

Thank you for your comment. We included this in Ethical approval statement in the revised manuscript in the following manner “The analysis displayed in the paper is based on the Ethiopian Demographic Health Survey-2016 which is a publicly available dataset with no identifiable information on the study members. The IRB-approved procedures for DHS public-use datasets do not in any way allow respondents, households, or sample communities to be identified. There are no names of individuals or household addresses in the data files. The geographic identifiers only go down to the regional level (where regions are typically very large geographical areas encompassing several states/provinces). Each EA (primary sampling unit) has a number in the data file, but their numbers do not have any labels to indicate their names or locations. The detail of the ethical issues has been published in the 2016 EDHS final report [10]. All the ethical concerns, including informed consent, are entirely followed in the EDHS-2016. Given these, no ethical approval or informed consent was required for the current study.” Dear reviewer 1, we also attached the approval letter from DHS (please see supplementary file) for your consideration. With all respect, thank you. 

Comment 2

2. Vast majority of the mother 60.3% had no formal education, so it would be great to include whether any association of unsafe child feces disposal practices with mother’s education.

Response 2:

Thank you for your wise advice. In Table 3 multilevel binary logistic regression analysis mother’s educational level associated with unsafe child stool disposal (p<0.05). The unadjusted crude odds ratio suggested that unsafe disposal of children’s stools was lower among educated mother than mothers with no education. However, as shown in Table 6 multilevel multivariable logistic regression analysis (model that include Individual- and community-level variables) mother’s educational level does not show any association with unsafe child stool disposal. Please see the table below. We also showed in the revised manuscript file. Thank you.

(Please observe the attached "Response to Reviewers" word doc for all TABLES) 

Comment 3

3. Having the toilet facilities is important to promote safe disposal, however if any household don’t have latrine how you considered in this analysis?

Response 3:

Thank you, our respected reviewer for your comment. Absolutely true “Having the toilet facilities is important to promote safe disposal”. In EDHS-2016 household sanitation facilities were categorized as 

Improved facility 

1. Flush/pour flush to piped sewer system 

2. Flush/pour flush to septic tank 

3. Flush/pour flush to pit latrine 

4. Ventilated improved pit (VIP) latrine 

5. Pit latrine with slab 

6. Composting toilet

Unimproved facility 

1. Flush/pour flush not to sewer/septic tank/pit latrine 

2. Pit latrine without slab/open pit 

3. Hanging toilet/hanging latrine 

4. Open defecation (no facility/bush/field) 

We follow similar procedure while categorizing sanitation facility (i.e., improved and unimproved facility). In our analysis, any household that doesn’t have latrine or those practice open defecation were categorized under unimproved sanitation facility. To clarify this thing, we include the following statement in footnote of Table 2. “Facilities that would be considered improved if any of the following types: flush/pour flush toilets to piped sewer systems, septic tanks, and pit latrines; ventilated improved pit (VIP) latrines; pit latrines with slabs; and composting toilets. Other facilities including households with no facility or use bush/field were considered unimproved.” Please see the revised manuscript. 

Our respected reviewer 1

Thank you for this is prestigious learning opportunity. With all respect. 

Response to reviewer’s comment 

For reviewer # 2

Dear respected reviewer 2

We are so happy to have your advice, comments and suggestion and we learn from your comments. Thank you for this opportunity. Please follow a point-by-point response to the reviewer’s comment below. We used a “Text Highlight Yellow Color” for all affected revisions and corrections in the “'Revised Manuscript with Track Changes” File. If there are any comments that need corrections, we are so happy to learn from you. With all respect, thank you.

Reviewer #2: Sahiledengle and colleagues mapped the distribution and described the risk of unsafe disposal of child stool in Ethiopia. Here, the study importance, objectives, and methods are clearly stated.

Comment 1

Minor comments:

1) What would happen if you use a continuous age variable rather than an age group in the model? What are the basis for participants’ age categorization?

Response 1:

Thank you, our respected reviewer. The choice of age variable categorization was guided by the previous works of literature (few examples are listed below) and we are not treating age as continues variable in the model in order to make similar comparison with related studies. 

1. Sahiledengle B. Prevalence and associated factors of safe and improved infant and young children stool disposal in Ethiopia: evidence from demographic and health survey. BMC Public Health. 2019;19(1):970. https://doi.org/10.1186/s12889-019-7325-9

2. Bawankule R, Singh A, Kumar K, Pedgaonkar S. Disposal of children’s stools and its association with childhood diarrhea in India. BMC Public Health. 2017:17(12). https://doi.org/10.1186/s12889-016-3948-2

3. Nkoka O. Correlates of appropriate disposal of children’s stools in Malawi: a multilevel analysis. BMC Public Health. 2020;20:604. https://doi.org/10.1186/s12889-020-08725-2

4. We also checked how the EDHS-2016 age of child was categorized 

In the previous works of literature, for instance a study from Malawi revealed that women whose children were aged 6–11 months (AOR: 3.06; 95% CI: 2.52–3.72), 12– 17 months (AOR: 6.81; 95% CI: 5.39–8.60), and 18–23 months (AOR: 6.58; 95% CI: 5.18–8.35) were more likely to dispose of their children’s stools compared with those whose children were aged < 6 months. Different studies, identified that unsafe child feces disposal is more prevalent among households those with younger children. In many cases, the reference category was children aged ≤ 2 years old. In recent DHS data including the Ethiopian DHS-2016 child stool disposal was collected for the youngest child under age 2 living with the mother. As children have similar characteristic with respect to child stool disposal we categorized age of the children in the following manner (< 6 month, 6-11 month, 12-17 month, and 18-23 months). The following assumptions were also used as a base while we categorizing age

• Children aged less than 6 months dependent on mothers and they used Dipper in many case; and it is difficult for them either to use potty or toilet- because of their age and stage of physical development. For this reason, unsafe child stool disposal is more prevalence at this stage. 

• Between 6 months to 11 months of age, babies grow and develop at an astounding rate. They may learn how to use potty. At this stage many start componentry feeding and their stool become odors and the potty training started by mothers/caregivers in Ethiopia. 

• 12-17 months: babies grow quickly they roll over, sit up, pick objects up, crawl, and some may even start use of potty by themselves. In many cases, use of potty/toilet training was well established.

• 18-23 months: at this stage and above years children are increasingly likely to use a toilet or latrine (if that is safe for child) /or potty themselves. Even at this stage they identify stool as waste. 

Comment 2

2) Include or explain in the supplementary or elsewhere the basis of categorizing the independent variables (e.g., wealth index, community poverty, etc.) to understand how you define the cut off and make sense to some of the nominal and dichotomous variables.

Response 2:

Thank you, our respected reviewer. We are so glad to have your comments for the improvement of our manuscript. As per your wise advice we include a supplementary file that explain the basis of categorizing the independent variables. Please see supplementary File 1.

Comment 3:

3) Table 4 is not informative.

Response 3:

Thank you, our respected reviewer. We revised Table 4 in order to make it more informative. We removed the lists of enumeration areas(clusters) detected for high unsafe child stool disposal in Ethiopia and supply a supplementary file that include the enumeration areas(clusters) detected for high unsafe child stool disposal with their coordinates for anyone who interested to see. Please see the revised manuscript table 4 and supplementary file 1. 

Comment 4:

4) I wonder if you could include or overlap data on the spatial prevalence of diarrhoea with spatial distribution of stool disposal. Their association is a strong basis for public health policy.

Response 4:

Thank you, our respected reviewer for your suggestion. In fact, there is a recent study that showed the spatial distribution of diarrhea in Ethiopia for three EDHS (2005-2016). 

1. Bogale, G.G., Gelaye, K.A., Degefie, D.T. et al. Spatial patterns of childhood diarrhea in Ethiopia: data from Ethiopian demographic and health surveys (2000, 2005, and 2011). BMC Infect Dis 17, 426 (2017). https://doi.org/10.1186/s12879-017-2504-8

As per your suggestion, we try to show the spatial prevalence of diarrhea with spatial distribution of stool disposal. Since, the coordinates for both diarrhea and unsafe child stool disposal are similar, we end up with overlapped map image that does not clearly show the spatial distribution for diarrhea and unsafe child stool disposal in clear manner. For this reason, we construct two images that show the spatial distribution of diarrhea and unsafe child stool disposal in Ethiopia in order to help the read to identify significant hot spot areas. Accordingly, we included the following information in the result section, “In Fig 4, the exploratory visualization of the spatial distribution diarrhea and unsafe child stool disposal were indicated. The highest proportions of diarrhea were observed in SNNPR region (Southern Ethiopia), while the highest hotspot areas of unsafe child stool disposal were detected in Tigray region (Northern Ethiopia).” Please see the revised manuscript result section and Figure 4. Thank you. 

Our respected reviewer 2

Thank you for this is prestigious learning opportunity. With all respect. 

Response to reviewer’s comment 

For reviewer #3

Dear respected reviewer 3

Our respected reviewer 3, thank you for this learning opportunity. We are so happy to have your advice, comments, corrections and suggestion. We are also so glad to see our paper improved because of your comment and wise advice. We learn several things from your comment and it help us to work hard in future. Dear reviewer 3, please follow a point-by-point response to the reviewer’s comment below. We used a “Text Highlight Yellow Color” for all affected revisions and corrections in the “'Revised Manuscript with Track Changes” File. If there are any comments that needs our correction, we are so happy to learn from you. With all respect, thank you.

Reviewer #3: This is a very important and interesting research. The introduction is sound and well structured. The rationale and objectives of the study are clearly spelt out. The methodology section is well written but needs improvement. Authors need to consider the introduction of new subsections, and provide more clarity on the categorization of variables. The data management and analysis plan was described appropriately. The data analysis section is robust, and the authors have done a very great job at pulling all the data together. However, there are issues with results presentation and description within text for a particular table. In addition, concerns on why the authors decided to pull the administrative regions together under agarian, pastoralist and city categories before the multivariate logistic analysis needs to be rechecked. The discussion is sound and well written. More specific comments have been raised in the additional file attached and authors should consider working on them.

Manuscript Title:

Geographical variation and factors associated with unsafe child stool disposal in Ethiopia: A spatial and multilevel analysis

Reviewer’s decision

The manuscript is of high quality considering the amount of work-done and analysis made. It is also well written. However, some minor corrections needs to be made prior and returned for another round of review before consideration for publication.

Summary of Reviewer’s comment:

This is a very important and interesting research. The introduction is sound and well structured. The rationale and objectives of the study are clearly spelt out. The methodology section is well written but needs improvement. Authors need to consider the introduction of new subsections, and provide more clarity on the categorization of variables. The data management and analysis plan was described appropriately. The data analysis section is robust, and the authors have done a very great job at pulling all the data together. However, there are issues with results presentation and description within text for a particular table. In addition, concerns on why the authors decided to pull the administrative regions together under agarian, pastoralist and city categories before the multivariate logistic analysis needs to be rechecked. The discussion is sound and well written. More specific comments have been raised in the additional file attached and authors should consider working on them. 

Response:

Thank you, our respected reviewer for your comment. We closely look all your concerns and comments and we addressed each point-by-point line by line in the revised manuscript. Please follow point by point response. With all respect. 

Comment 1

Reviewer’s Comment

TOPIC:

The title of this manuscript is appropriate and concise, 

Response 1:

Thank you for this encouraging words. 

Comment 2:

INTRODUCTION 

Line 48: recast as..”….open fields, garbages, drainages and burying in soils are considered unsafe as it exposes vulnerable children who interact with such to many fecal-oral diseases.

Response 2:

Thank you, our respected reviewer for your helpful comment. As per your wise advice we correct the first line of the introduction section accordingly. Please see the revised manuscript introduction section. 

Comment 3:

Line 50: child stool disposal is..

Response 3:

Thank you, our respected reviewer for your comment. As per your wise advice we correct accordingly. Please see the revised manuscript section. 

Comment 4:

Line 54: Recast as “Futhermore, there are established evidences on the effect …….

Response 4:

Thank you, our respected reviewer for your comment. As per your wise advice we recast the stated sentence accordingly. Please see the revised manuscript section. 

Comment 5:

Line 58: Replace “with available” with “according to”

Response 5:

Thank you, our respected reviewer for your comment. As per your wise advice we the replace “with available” with “according to”. Please see the revised manuscript section. 

Comment 6:

Line 58: It is not uncommon that major contributors….

Response 6:

Thank you, our respected reviewer for your comment. We revised the whole sentence following your wise advice. Please see the revised manuscript section. 

Comment 7:

Line 61-62: Move this to Line 58 after the reference [2]

Response 7:

Thank you, our respected reviewer for your suggestion. We follow your advice and we revised the introduction section accordingly and we put line 58 immediately after reference number 2. It readd as “In Ethiopia, hygienic child stool management could be a tremendous challenge and putting the nation among the most noticeably awful third of 38 African nations for the percentage of children whose feces are safely disposed of according to the Multiple Indicator Cluster Survey (MICS) [2]. According to a later pooled information from the Ethiopian DHS (2000-2016), 77 percent of children’s feces disposed of unsafely [10].” Thank you. Please see the revised manuscript section. 

Comment 8:

Line 63: previous studies has identified…

Response 8:

Thank you, our respected reviewer for your comment. Comment accepted and we correct accordingly. Please see the revised manuscript section. 

Comment 9:

Line 69-70: Are the authors trying to discuss the findings of the research ahead of time. If yes, please expunge. 

Response 9:

Thank you, our respected reviewer for your comment. We apologized for this mistake. As per your wise advice we removed the stated sentence from the introduction section. Please see the revised manuscript section. 

Materials and Methods 

Comment 10:

Authors should restructure their methods section. It is expected that, the authors would provide a as a starting line, a brief summary of the study location in this case, Ethiopia i.e the districts, populations, health system, language, etc. Then an additional sub-section on study design should be developed. This is where the authors would explain the type of design employed, as it appears a secondary analysis was done.

Authors can then follow up with another section of data source and extraction, where they will provide very explicit details on how data were sourced, and explain in great details the EDHS 

Response 10:

Thank you for your comment. As per your wise advice we followed the suggested outline. We provide a brief summary of the study area, then we describe the study design, and finally we provide explicit details on data source and sampling issues of the EDHS. Please see the revised manuscript materials and methods section. Thank you. 

Comment 11:

The sections on study variables is fine.

Response 11:

Thank you, our respected reviewer. 

Comment 12:

Line 114: rephrase as “was guided by already existing literatures. 

Response 12:

Thank you, our respected reviewer for your comment. We corrected and rephrase accordingly. Please see the revised manuscript. 

Comment 13:

Line 119: replace recorded with “categorized”

Response 13:

Thank you, our respected reviewer for your comment. We corrected accordingly. Please see the revised manuscript. 

Comment 14:

Line 120: replace combined to form the with “categorized as”

Response 14:

Thank you, our respected reviewer for your comment. We corrected accordingly. Please see the revised manuscript. 

Comment 15:

Line 121: Replace combined as the with “categorized as”

Response 15:

Thank you, our respected reviewer for your comment. We corrected accordingly. Please see the revised manuscript. 

Comment 16:

Line 122-123 (Table 1): Can authors please provide descriptive text on what metrics they used in estimating community poverty as low or high. It appears they used wealth index that has been described on this same table. How did they go about the grouping?

Response 16:

Thank you, our respected reviewer for your comment. As per your wise advice we briefly describe how the community poverty level was constructed in the independent variable section as well as in supplementary file 1. In brief, community poverty level was generated by aggregating the individual characteristics in a cluster since EDHS did not collect data that can directly describe the characteristics of the clusters except the place of residence. The aggregate community poverty level was constructed by aggregating individual level characteristics at the community (cluster) level. Accordingly, community poverty level was an aggregate wealth index categorized as high or low, which is the proportion of women in the poorest and poorer quintile derived from data on wealth index which is categorized as low and high poverty community. Since the aggregate community poverty level value is not normally distributed, it was categorized into high and low groups based on the median value. Community poverty level was categorized as high if the proportion of women from the two lowest wealth quintiles in a given community was higher than the median value and low if the proportion was less than and equal to median value. Please see the revised manuscript independent variable section as well as in supplementary file 1. Thank you. 

Comment 17:

Table 1: Authors should provide more descriptive text on how the wealth index was estimated from the source survey. They should provide very clear notes that would guide the readers. Same thing applied to source of drinking water. How did they arrive at the category of improved and unimproved?.

Response 17:

Thank you, our respected reviewer, for this constructive comment. Following your wise suggestion, we provide a descriptive text on how the wealth index was estimated including source of drinking water was categorized in the foot not in Table 1. Please see the revised manuscript table 1 foot note as well as supplementary file 1 . Thank you. 

Comment 18:

Table 1: Can authors expunge the “categorized into that reflects on all the cell in the category column. I t would be more appropriate to just have the categories listed e.g (1) male (2) female.

Response 18:

Thank you, our respected reviewer, for this helpful comment. Following your wise suggestion we correct Table 1 accordingly. Please see the revised manuscript table 1. Thank you. 

Comment 19:

Line 125: should start with “Secondary data obtained were imported(?) and analyzed in…….

Response 19:

Thank you, our respected reviewer, for this helpful comment. We rephrase the data management and analysis section according to your wise advice. Please see the revised manuscript. Thank you. 

Comment 20:

Line 127-128: …weighting procedure has been sufficiently described in EDHS methodology [10].

Response 20:

Thank you for your comment, we correct accordingly. Please see the revised manuscript. 

Comment 21:

Line 129-131: Please recast as follow”………….children within a cluster are more similar………clusters. As such, a multilevel model is……..because it allows analysis based on hierarchical structure of variables”. 

Response 21:

Thank you for your comment, as per your wise advice we correct accordingly. Please see the revised manuscript. 

Comment 22:

Line 145: Remove “ STATA……analyses”.

Response 22:

Thank you for your comment, we correct accordingly. Please see the revised manuscript. 

RESULTS

Comment 23:

Table 2; Can authors explain and address why some of the demographic variables do no amount to 4145. If there are no response or missing data, they should be described so the readers can have a better understanding

Response 23

Thank you , our respected reviewer for comment. If fact all variables add to 4145. The main reason we see in total count less than 4145 in variables [Sex of the child (n=4144); Mother’s age (4143)] was because of sampling weight and some missing values. Following your wise advice, we include an asterisk indicates the reason why the total count less than 4145. Hear is unexampled for variable sex of the child. 

Please see the attached "Response to Reviewers" word doc for all TABLES.

Comment 24:

Table 2: Under mothers working status, can authors find a more befitting category rather than the Not working and working used 

Response 24:

Thank you. As per your wise advice we revised mother working status into “mother’s employment status” as per the DHS tool in the revised manuscript. Please see the revised manuscript and supplementary file 1 for detail description. 

Comment 25:

Table 2: Still very much bothered about the community poverty level here. It appears vague

Response 25:

Thank you, our respected reviewer for your constructive comment. As per your wise advice we clarify community poverty level in the independent variable section and supplementary file 1. In brief, community -level variables, such as community poverty level was generated by aggregating the individual characteristics in a cluster since EDHS did not collect data that can directly describe the characteristics of the clusters except the place of residence. The aggregate community poverty level was constructed by aggregating individual level characteristics at the community (cluster) level. Accordingly, community poverty level was an aggregate wealth index categorized as high or low, which is the proportion of women in the poorest and poorer quintile derived from data on wealth index which is categorized as low and high poverty community. Since the aggregate community poverty level value is not normally distributed, it was categorized into high and low groups based on the median value. Community poverty level was categorized as high if the proportion of women from the two lowest wealth quintiles in a given community was higher than the median value and low if the proportion was less than and equal to median value. 

Comment 26:

Table 3: This table has been poorly described. Authors should take their time to explain the details contained in the table. At a first glance, I believe the authors are focusing on the odds of practicing unsafe stool disposal. The results presented here shows varying levels of odds, which I found quite interesting. For instance, the odds of practicing unsafe disposal was reduced as the children increase in age, except for those aged 12-17 months. Looking at the data this way may provide very great insight on how to discuss the results subsequently. 

Response 26:

Thank you, our respected reviewer, for your interesting comment. Following your wise advice, we describe the detail contain of Table 3. In the following manner “As shown in Table 3, binary multilevel logistic regression analysis was used to present unadjusted OR (95% CI) for individual and community level variables to identify factors associated with unsafe child stool disposal. Individual level characteristics such as sex of the child, age of the child, diarrhea conditions of the child, mother educational level, mother’s employment status, household wealth index, toilet facility, and source of drinking water were significantly associated with unsafe child stool disposal at p<0.05 (Table 3). All the community level characteristics (region, place of residence, and community poverty level) were found to be significantly associated with unsafe child stool disposal at p<0.05 (Table 3).” As you understand, table 3 contains unadjusted or crude OR estimates, which limited us to explain the odds as you observe in the adjusted Table 3 (that showed adjusted OR results). Thank you for your comment. Please see the revised manuscript. 

Comment 27:

Line 182-197: Since authors knew they would still present and discuss their findings by districts/regions i.e Tigray, Amhara etc, why did they group these administrative regions together in their multivariate logistic model. It would have been great if they analyse them separately too, and have odd ratios that would add credence to the maps they presented as fig 1 and 2.

Response 27:

Thank you, our respected reviewer for your comments. Ethiopia has 11 administrative regions including 2 city administrations (namely Tigray, Amhara, Oromiya, SNNP, Gambella, Benshangul Gumuz, Somali, Afar, Harar, Addis Ababa, and Dire Dawa). However, contextually the country is categorized as agrarian, pastoralists, and city-based population. The main reason we follow this grouping of these administrative regions in multivariate logistic model was due 

1st: The interest of the current study was not in the regions delineated for administrative purposes, which might not necessarily be related to child stool disposal of the population.

2nd: lack of adequate cell to run multilevel logistic regression (please check two by two table showing weighted child stool disposal vs regions). Some of the regions like Afar, Gambela, Harari, and Dire Dawa) have less cell count which affected our estimates and wider the confidence interval as multilevel analysis required larger sample size. In such case, it is highly recommended to group similar cells to find a robust finding. 

3rd: Most importantly, region was a category with more than 5 categories: In logistic regression analysis the independent variable category should be more than 5 categories as a general guiding principle (in our case Ethiopia regions were categorized in 11 administrative sections), which is more than the expected category. In such case, it is highly advisable to regroup the category into 5 or less category in order to have stable estimates. In run running logistic regression analysis having independent variables with more than 5 categories the model considered the recent category as continuous variable. In order to resolve this regrouping, the category in less than 5 group is crucial. 

Comment 28

Table 4: Can the column on enumeration areas be removed. At the moment it adds no value to the table, it would be better if the clusters can be provided as a supplementary file, with aadditional descriptive lines provided. But for this present table, the authors should retain 7 columns. The first can be titled category of clusters i.e primary, secondary etc. while the second column, can be titled number of clusters detected i.e. 201, 26 ..

Response 28:

Thank you, our respected reviewer. As per your wise advice we revised Table 4 accordingly and we include a supplementary fil, with additional information on list of enumeration areas and their coordinates. Please see the revised manuscript table 4. 

Comment 29

Line 211-217: The footnotes provided here should rather be converted to main text, and follow with line 207.

Response 29:

Thank you, our respected reviewer. Following your wise advice, we converted the footnotes provided in Table 4 to main text. Please see the revised manuscript spatial scan statistical analysis section. Thank you.

Comment 30

Line 224-225: Authors should check the write up. The 39.61% is from the null model not community level factors

Response 30:

Thank you, our respected reviewer. In short yes, the ICC (39.61%) is from the null model not community level factors; meaning in the null model, significant variation in unsafe child stool disposal was observed among mothers across communities was observed with an ICC of 39.61 % justifying the use of multilevel analysis approach (i.e., variation in terms of unsafe safe child stool disposal could be attributed to unobserved community characteristics). 

Comment 31

Line 225: Please recast as “Furthermore, between -cluster variability….

Response 31

Thank you, our respected reviewer. We recast the sentence as per your wise advice. Please see the revised manuscript. 

Comment 32

Table 5: The heading row shoes AOR (95% CI). Please authors should take this out

Response 32

Thank you, our respected reviewer. We remove AOR (95%CI) from Table 5 heading rows as they are not necessary. Thank you. Please see the revised manuscript table 5. 

Comment 33

Table 6: Looks great. 

Response 33

Thank you, our respected reviewer for your support and wise advice. 

Comment 34

Line 251: insert a parenthesis after middle, before AOR.

Response 34

Thank you, our respected reviewer. We insert a parenthesis and we correct our mistake according. Thank you. Please see the revised manuscript. 

DISCUSSION

Comment 35

The discussion section is sound, and appropriately written in context of the results presented

Response 35:

Thank you, our respect reviewer for your advice and support. 

Limitations

Comment 36

Line 335: Start with Although,

Response 36:

Thank you, our respect reviewer. As per your wise advice we correct accordingly. Please see the revised manuscript limitation section. 

Comment 37

Line 336: expunge “, a model”

Response 37:

Thank you, our respect reviewer. As per your wise advice we correct accordingly. Please see the revised manuscript limitation section. 

Comment 38

Line 341: expunge, in

Response 38:

Thank you, our respect reviewer. As per your wise advice we correct accordingly. Please see the revised manuscript limitation section. 

Comment 39

Line 343: recast as “…..nature of the survey is not appropriate to estimate the cause and effect…….

Response 39:

Thank you, our respect reviewer. As per your wise advice we correct accordingly. Please see the revised manuscript limitation section. 

Our respected reviewer 3

Thank you for this is prestigious learning opportunity. With all respect.

---

## [Editor Report · Decision Letter 1]

15 Apr 2021

Geographical variation and factors associated with unsafe child stool disposal in Ethiopia: A spatial and multilevel analysis

PONE-D-20-24200R1

Dear Dr. Sahiledengle,

We’re pleased to inform you that your manuscript has been judged scientifically suitable for publication and will be formally accepted for publication once it meets all outstanding technical requirements.

Kind regards,

Harvie P. Portugaliza, Ph.D.

Guest Editor

PLOS ONE

Additional Editor Comments (optional):

In the revised manuscript, the authors addressed minor corrections and essential revisions suggested by three reviewers; for example, indicating the association of unsafe child feces disposal practices with mother’s education (Reviewer 1), clarifying in the text the basis for variable categorization, and presenting the results more clearly (Reviewer 2 and 3).
---

## [Editor Report · Acceptance letter]

19 Apr 2021

PONE-D-20-24200R1 

Geographical variation and factors associated with unsafe child stool disposal in Ethiopia: A spatial and multilevel analysis 

Dear Dr. Sahiledengle:

I'm pleased to inform you that your manuscript has been deemed suitable for publication in PLOS ONE. Congratulations! Your manuscript is now with our production department. 

Kind regards, 

on behalf of

Dr. Harvie P. Portugaliza 

Guest Editor

PLOS ONE